# Gill Transcriptome, Proteome, and Histology in Female *Eriocheir sinensis* Under Copper Stress

**DOI:** 10.3390/ijms26104711

**Published:** 2025-05-14

**Authors:** Tingshuang Pan, Tong Li, Min Yang, Chengchen Yu, He Jiang, Jun Ling

**Affiliations:** 1Fishery Institute of Anhui Academy of Agricultural Sciences, Hefei 230031, China; pantingshuang@aaas.org.cn (T.P.); litong@aaas.org.cn (T.L.); yangmin@aaas.org.cn (M.Y.); yuchengchen@aaas.org.cn (C.Y.); jianghe@aaas.org.cn (H.J.); 2Key Laboratory of Aquaculture & Stock Enhancement in Anhui Province, Hefei 230031, China

**Keywords:** gene, protein, histology, Chinese mitten crab

## Abstract

Cu is a chemical contaminant that is toxic to aquatic animals at certain concentrations. The present study describes the gill transcriptome, proteome, and histology of the Chinese mitten crab (*Eriocheir sinensis*) subjected to copper stress. Female 14-month-old *E. sinensis* (n = 60) crabs (79.6 ± 4.8 g, body weight) were randomly divided into two groups and subjected to copper stress at concentrations of 0 μg/L (Blank group, GBL) and 50 μg/L (Copper group, GCP) for 96 h. In total, 278 upregulated and 189 downregulated differentially expressed genes (DEGs) were identified in the GBL and GCP groups. In addition, upregulated and downregulated differentially expressed proteins (DEPs) in the GBL and GCP groups were 260 and 308, respectively. An integrated analysis demonstrated that the three DEGs overlapped between the two omics approaches. Comparative omics analysis indicated that seven GO terms were significantly (*p* < 0.05) enriched by overlapping DEGs in the transcriptome and proteome. Further analysis revealed that only one overlapping DEG (stumps) was enriched in two common KEGG pathways, the PI3K-Akt and B cell receptor signaling pathways. Histological analyses showed that copper-stressed gills had collapsed lamellae with enlarged marginal vessels and shortened interlamellar spaces due to the disruption of the pillar cells and cuticles. These results demonstrate the variations in copper-stressed gills and will be helpful for better understanding the mechanisms of copper toxicity in *E. sinensis*.

## 1. Introduction

Copper (Cu) is an important micronutrient in living organisms. It plays a key role in many metabolic processes, including mitochondrial function, electron transport, oxygen transport, and respiration [1]. However, when copper exceeds the optimal levels, cells are damaged and pathways related to immunology are elevated [2]. Cu in aquatic systems accumulates from both anthropogenic and natural sources [3]. Copper sulfate is useful for controlling parasitic and bacterial diseases in aquatic animals, cyanobacterial blooms, and filamentous algae in reservoirs [4,5]. The continuous use of copper sulfate has resulted in copper accumulation in aquatic systems. The upper limit of copper as prescribed by the Chinese National Water Quality Standard for Fisheries (GB11607-89) is ≤0.01 mg/L [6]. Copper concentration exceeding 0.01 mg/L is regarded as abnormal according to this standard.

*Eriocheir sinensis* is an economically important aquatic animal cultured in China. Copper may disrupt the normal physiological processes of organisms when it exceeds the normal physiological range. Recently, the negative effects of high Cu concentrations have been studied in aquatic animals [2,7,8,9]. When *E. sinensis* was treated with copper (50 μg/L) for 96 h, lipid metabolism was affected, including lipid synthesis and triglycerides hydrolysis increased, and fatty acid β-oxidation decreased [8]. Additionally, antioxidative capacity decreased; lipid peroxidation was promoted in the muscle, gill, hepatopancreas, and hemolymph; and genes related to autophagy were upregulated when *E. sinensis* was stressed with copper for 5 days [9]. Tang et al. reported that gill cells were damaged, and apoptosis and oxidative stress processes were induced by copper after being subjected to copper stress for one day [2]. Malondialdehyde increased, expression of anti-apoptotic and immune-related genes decreased, and the antioxidant enzymes were suppressed when *E. sinensis* was exposed to copper (40 μg/L) for 7 weeks [10]. Bioassays of acute short- and long-term copper toxicity in brown bullhead fish (*Ictalurus nebulosus*) revealed the occurrence of morphological changes in the skin, liver, and gills during histomorphological and histochemical analyses [11].

Gills play important roles in respiration, osmoregulation, excretion, and ion transport in aquatic animals [12]. Compared to other organs, the gills are directly exposed to water and accumulate more metal pollutants [13]. Environmental stress negatively affects crustacean metabolism. When *E. sinensis* and *Palaemonetes sinensis* are stressed by hypoxia, lactate hydrogenase and succinate dehydrogenase in their gills are significantly affected [14]. Treatment of crustaceans with high levels of copper results in the inhibition of osmoregulation enzymes, ion transport, ammonia excretion, and gas exchange in the gills [15]. When *Macrobrachium nipponense* was treated with different concentrations of nanoplastics, ATPase activity decreased and the expression of ion-transport-related genes was first upregulated and then downregulated in the gills [16].

Transcriptomes and proteomes have been used to analyze DEGs and DEPs during environmental pollutant stress [17,18]. When *Epinephelus coioides* was exposed to CuSO_4_ or CuNPs for 24 h, the regulation of some DEGs and DEPs was significantly altered, and the common DEGs and DEPs in the two treatments were 911 and 75, respectively [17]. To investigate copper toxicity in *Gymnocypris eckloni*, the transcriptome was used to analyze the changes in different tissues, and the results showed that DEGs dramatically changed in the liver and gills [18].

To detect variations in the gills, the transcriptomes and proteomes of copper-stressed *E. sinensis* were studied. The DEGs and DEPs of the gill transcriptomes and proteomes from the GCP and GBL groups were analyzed. Combined analysis using the two omics approaches identified the common DEGs, DEPs, GO terms, and KEGG pathways. Histological variations in the gills were also analyzed. These results will help clarify the mechanism of copper stress and provide valuable genomic, proteomic, and histological information for future studies on *E. sinensis*.

## 2. Results

### 2.1. Transcriptome and Proteome Annotation

A total of 241.35 M and 235.66 M raw reads of transcriptome were achieved from the GBL and GCP groups. The clean reads were 236.92 M and 230.56 M, and GC contents were 46.98% and 46.53% in the GBL and GCP groups. The ratios of the mapped reads were 66.75% and 72.29% for the GBL and GCP groups, respectively (Table 1). Raw transcriptome data (PRJNA1178304) were deposited in the NCBI SRA database.

Gill proteomes from the GBL and GCP groups were sequenced using data-independent acquisition (DIA). In total, 13,285 unique peptides and 5904 proteins were obtained from the GBL and GCP groups, respectively. The proteins were mostly distributed in 10–100 kDa (Figure 1). Raw proteomic data (OMIX007550) were obtained from the China National Center for Bioinformation.

### 2.2. Tanscriptome and Proteome Analysis

PCA analyses showed 66.37% and 41.2% variations at the transcriptome (Figure 2A) and proteome levels (Figure 2B), respectively.

In total, 467 DEGs, including 278 upregulated and 189 downregulated genes, were identified (Appendix A). Moreover, 568 DEPs, including 260 upregulated and 308 downregulated ones, were identified (Appendix A).

The correlation between the transcriptome and proteome was analyzed in the GBL and GCP groups. A total of 1582 genes (Appendix A) were correlated with proteins, including 140 genes (Appendix A) that were significantly (*p* < 0.05) correlated with proteins between the two omics. Among all the significantly correlated (*p* < 0.05) genes, 124 were positively correlated, and 16 were negatively correlated. The coefficient of correlation between the transcriptomes and proteomes was 0.382 (Figure 3).

### 2.3. DEGs and DEPs Analysis

In the biological process (BP), the most enriched DEGs were protein folding, male germline stem cell population maintenance, eggshell chorion assembly, carbohydrate biosynthetic process, hydrogen peroxide metabolic process, prostaglandin biosynthetic process, proteolysis, and chorion-containing eggshell formation. In the cellular component (CC), most DEGs were enriched in the collagen trimer, oligosaccharyl transferase complex, extracellular matrix, and perinuclear endoplasmic reticulum. In molecular function (MF), most DEGs were enriched in DDT-dehydrochlorinase activity, unfolded protein binding, peptide binding, coupled ATPase activity, glutathione transferase activity, signaling pattern recognition receptor activity, structural constituents of the cuticle, and prostaglandin–endoperoxide synthase activity (Figure 4A).

In the BP subcategory, most DEPs were enriched in the immune response, RNA export from the nucleus, intracellular protein transport, positive regulation of apoptotic signaling pathway, protein methylation, endosome organization, and attachment of GPI anchor to protein. In the CC subcategory, most DEPs were enriched in the cytoplasm, cytoskeleton, glucosidase II complex, nuclear pores, HOPS complex, GPI-anchor transamidase complex, adherens junction, and catenin complex. In the MF subcategory, most DEPs were enriched for pseudouridine synthase activity, diacylglycerol-dependent serine/threonine kinase activity, D-serine ammonia–lyase activity, oxidoreductase activity, protein domain-specific binding, structural constituents of nuclear pores, protein serine/threonine phosphatase activity, actin filament binding, and iron ion binding (Figure 4B).

Three DEGs overlapped between the two omics approaches (Table 2). Two DEGs (LOC126994049, stumps) were significantly (*p* < 0.05) up-regulated in the transcriptome level and were downregulated at the proteome level. One DEG (SACS) was significantly (*p* < 0.05) downregulated in the transcriptome and upregulated in the proteome.

Comparative omics analysis indicated that seven GO terms were significantly enriched by overlapping DEGs in the transcriptome and proteome, including regulation of the inflammatory response, positive regulation of phosphatidylinositol 3-kinase signaling, DNA replication initiation, positive regulation of protein kinase B signaling, meiotic cell cycle, DNA repair, and low-density lipoprotein particle receptor binding (Table 3).

DEGs from the GCP and GBL groups were subjected to KEGG pathway analysis and classified into 173 pathways, among which 13 pathways were significantly enriched. As shown in Figure 5A (Appendix A), the most enriched pathways included hematopoietic cell lineage, drug metabolism (cytochrome P450, fructose, and mannose metabolism), xenobiotic metabolism by cytochrome P450, insect hormone biosynthesis, hepatocellular carcinoma, arachidonic acid metabolism, prion diseases, fluid shear stress and atherosclerosis, B cell receptor signaling pathway, longevity regulating pathway–worm, and primary immunodeficiency.

The DEPs between the GCP and GBL groups were used for KEGG pathway analyses, and 187 pathways were identified. Among these, six were significantly enriched. The 30 most enriched pathways are shown in Figure 5B (Appendix A). The most enriched pathways include bacterial invasion of epithelial cells, autophagy–yeast, arrhythmogenic right ventricular cardiomyopathy, valine leucine and isoleucine degradation, Leukocyte transendothelial migration, glycosylphosphatidylinositol (GPI)-anchor biosynthesis, maturity onset diabetes of the young, butanoate metabolism, pinene camphor and geraniol degradation, cytoskeleton in muscle cells, spliceosome, Nucleocytoplasmic transport, phenylalanine metabolism, caprolactam degradation, adherens junction, RNA polymerase, and endometrial cancer.

Further analysis revealed that only one overlapping DEGs (stump) was enriched in the two common pathways of the transcriptome and proteome, including the B cell receptor and PI3K-Akt signaling pathway (Table 4).

### 2.4. Histological Analysis

The gills in the GBL group had normal cuticles (CU), pillar cells (PC), marginal vessels (MV), and interlamellar spaces (ILS) (Figure 6A). In contrast, the gills in the GCP group had collapsed lamellae with an enlarged marginal vessel and shortened interlamellar space due to disruption of the pillar cells and cuticle (Figure 6B).

## 3. Discussion

Excess Cu is toxic to aquatic animals, and its toxic effects are affected by its concentration and exposure time [19]. SOD activity showed a downward trend in gills after Cu^2+^ exposure (0, 0.04 mg/L, 0.18 mg/L, 0.70 mg/L) for 5 days and was strongly decreased in crabs exposed to 0.70 mg/L of Cu^2+^ [8]. In the gills of Gymnocypris eckloni, the expression of GeHsp90 was significantly upregulated and significantly downregulated after copper stress and then recovered to normal levels [20]. When Rhamdia quelen was subjected to stress at three copper concentrations (2, 7, and 11 µg/L) for 96 h, leukocyte infiltration and areas of necrosis causing raised levels of lesions were observed upon exposure to copper at 7 and 11 µg/L [19]. Similarly, copper stress (50 μg/L) in E.sinensis caused an inflammation response and gill damage in the present study. Cu exposure also altered immune-related pathways in E.sinensis, affecting genes and processes such as the B cell receptor and the PI3K-Akt signaling pathway.

### 3.1. Common DEGs and DEPs in Two Omics

The effects of Cu exposure on immunity, lipid metabolism, and oxidative stress have been reported in *E. sinensis* [2,7,8,9]. When *E. sinensis* was exposed to Cu^2+^ for 24 h, 2486 DEGs were identified, and GO functional analysis revealed that most DEGs were related to antioxidation, detoxification, and immunity [2]. When *E. sinensis* was exposed to Cu^2+^ (50 μg/L) for four days, the expression of genes associated with lipid synthesis (ACC, FAS, SCD) was considerably reduced after a decline in SREBP expression [8]. In the present study, five genes were downregulated and two were upregulated in the lipid metabolism pathway, indicating that lipid metabolism was inhibited under copper stress. When *E. sinensis* was exposed to four different Cu^2+^ concentrations (0, 0.04, 0.18, 0.70 mg/L) for five days, the antioxidative capacity decreased, and lipid peroxidation increased with the concentration of Cu^2+^ elevated in the gills, hepatopancreas, hemolymph, and muscles [9]. However, an integrated analysis of the gill transcriptome and proteome in *E. sinensis* caused by Cu^2+^ has not been reported.

A total of 467 DEGs and 568 DEPs were identified between the two groups. Further analysis showed 278 upregulated and 189 downregulated DEGs, and 260 upregulated and 308 downregulated DEPs between the two groups. More DEGs were upregulated and more DEPs were downregulated when *E. sinensis* was stressed with Cu for a short time. The proteome PCA results (Figure 2B) indicated that the samples in the GBL and GCP groups could be separated. The coefficient of correlation between the transcriptomes and proteomes was 0.382, which is not high. This may be affected by the transcription of DNA into mRNA and translation of mRNA into protein by different factors, such as the regulation of transcription, translation, and post-translation. These changes lead to changes in mRNA transcript and protein numbers, resulting in a low correlation between the transcriptome and proteome. All three genes (SACS, stumps, LOC126994049) showed opposite expression trends at the gene and protein levels. These opposing expression trends in the transcriptomes and proteomes of the GCP and GBL groups may have been influenced by post-transcriptional modifications.

### 3.2. GO Terms and KEGGs

In the present study, seven GO terms were significantly enriched by overlapped DEGs in transcriptome and proteome from the GCP and GBL groups, including DNA replication initiation, meiotic cell cycle, and DNA repair, etc., which showed that these GO terms played important roles in Chinese mitten crabs exposed to copper stress. This indicated that these GO terms played important roles in *E. sinensis* under copper stress.

Inflammation is indicative of an immune response to infection or damage [21]. Inflammation caused by Cu stress has been reported in both animals and humans [22]. Copper-stress-regulated cytokines increase or decrease via different pathways [23]. “Inflammatory response” was identified as a significantly enriched (*p* < 0.05) GO term, and the degree of enrichment was 4.42 according to the combined omics analysis, indicating that, in the present study, an inflammatory response was activated in the gills of *E. sinensis* by copper stress. Feng [9] reported that an inflammatory response was initiated in the hepatopancreas of Chinese mitten crabs after copper exposure to copper. When Chinese mitten crabs were treated with copper for seven weeks, the expression of inflammation-related genes increased, and the activity of damage-related enzymes was elevated [10].

Copper can potentially generate reactive oxygen species (ROS) and damage macromolecules such as DNA [24]. DNA damage can cause mutations and result in chronic disease [25]. DNA repair systems can reduce the occurrence of mutations and chromosomal aberrations. In the present study, “DNA repair” was one of the most significantly enriched GO terms, and the degree of enrichment was 0.66 in the GCP group, indicating that Cu inhibited DNA repair in the gills. When zebrafish (*Danio rerio*) were exposed to a 20 mg/L Cu solution, in hepatocytes, most genes involved in DNA repair were inhibited [26].

A previous study also showed that when human cell extracts and purified recombinant proteins were treated with micromolar copper concentrations, the activity of kinase and phosphatase in polynucleotide kinases was inhibited [27]. There are two common KEGG pathways associated with the transcriptome and proteome in the present study. The KEGG pathway results showed that the B cell receptor and PI3K-Akt play important roles in the copper stimulus. The phosphoinositide3-kinase (PI3K)/Akt signaling cascade is a major mediator of insulin signaling that regulates proliferation and apoptosis [28]. Cu^2+^ activates PI3K and Akt. Akt is activated by PI3K, resulting in the phosphorylation of Akt substrate phosphorylation [29].

### 3.3. Gill Histology

A histological analysis provides direct evidence of tissue damage and cellular alterations caused by the toxicants. In the present study, gill tissues of the control group exhibited normal architecture, whereas crabs exposed to the copper-stressed group (50 μg/L) revealed severe structural damage in the gills. Gills are the primary sites of nanoparticle entry into aquatic animals, and exhibit edema, lamellar epithelial lifting, cellular degeneration, and necrosis, all of which impair respiration and osmoregulation. Such alterations in the gill structure are consistent with the observations of previous studies, where nanoparticles have been shown to cause gill damage [30,31,32]. Edema is a frequent lesion in the gill epithelium when fish are stressed by heavy metals, which may result in osmotic imbalance [33].

## 4. Materials and Methods

### 4.1. Ethics

All crab experiments were conducted under the national regulations on laboratory animals of China and approved by the Experimental Animal Welfare and Ethical of Anhui Academy of Agricultural Sciences guidelines of use of animals for research (Approval Code: AAAS2022-20).

### 4.2. Reagents

CuSO_4_ 5H_2_O were purchased from Sangon Biotec Co., Ltd. (Shanghai, China).

### 4.3. Experimental Animals

Healthy female *E. sinensis* (79.6 ± 4.8 g, body weight) at the age of 14 months were from Fisheries Institute of Anhui Academy of Agricultural Sciences and acclimated in 10,000 L plastic tank. *E. sinensis* were fed with sinking pellets (Tongwei Feed Co., Ltd., Hefei, China) twice daily. The formula feed comprised 40% protein, 12.0% crude fiber, 4.0% lipids, 16.0% ash, 0.6% total phosphorus, and 2.2% lysine. A week later, 60 *E. sinensis* were transferred into six 200 L tanks (10 *E. sinensis* per tank) for exposure experiments. The culturing conditions were kept as follows: dissolved oxygen 8.1 ± 0.2 mg/L, temperature 27 ± 0.3 °C, and pH 7.4 ± 0.1.

### 4.4. Exposure Experiment and Sampling

Sixty *E. sinensis* were divided into two groups and exposed to 0 (Blank group, GBL) and 50 μg/L (Copper group, GCP) of the copper solution for 96 h as reported [8]. Each group had three replicates, with 30 *E. sinensis* in each group. Water was changed quarterly with the same copper concentration each day, and no feeds were provided. After 96 h exposure, ten *E. sinensis* were sampled randomly from each group, and the gills were immediately collected after anesthetization with an ice bath. In each group, five gills were stored in a −80 °C refrigerator for the omics analysis. The other gills were placed in 4% paraformaldehyde for the histology analysis.

### 4.5. Transcriptome Analysis

The transcriptome was sequenced using Illumina sequence platform. TRIzol reagent (Invitrogen, Waltham, MA, USA) was used to isolate total RNA from ten gills. The purity and amount of total RNA were assessed using a NanoDrop ND-1000 (NanoDrop, Waltham, MA, USA). Total RNA integrity was determined using an Agilent 2100 Bioanalyzer (Agilent Technologies, Palo Alto, CA, USA) and RNase-free agarose gel electrophoresis. Full-length cDNA was constructed using the TruSeq RNA Sample Prep Kit (Illumina, San Diego, CA, USA) according to the manufacturer’s instructions. The resulting cDNA library was sequenced using an Illumina Novaseq 6000 (Shanghai OE Biotech Co., Ltd., Shanghai, China), and 150-bp paired-end reads were generated. Fastp (version 0.18.0) [34] was used to filter the original data containing adapters or low-quality base (*q* value ≤ 20) exceeding 50%. Clean reads were used to map Chinese mitten crab reference genome (NCBI_ ASM2467909v1). The mapped reads were assembled by StringTie with the default parameters. HISAT2 [35] was used to perform the expression level. PCA was performed using the value of the relative difference in the transcriptomes to separate the HS and NS groups. DEGs were selected by DESeq [36] with *q* < 0.05 and Foldchange > 2 or Foldchange < 0.05. GO and Kyoto Encyclopedia of Genes and Genomes (KEGG) pathways enrichment were calculated by R (v 3.2.0). GO terms or KEGG pathways with *q* value < 0.05 were significant.

### 4.6. Proteome Analysis

The DIA proteome method was used to analyze the gill of female Chinese mitten crabs. Total protein in the gills was extracted as Pan [37] reported. Protein content was determined using the bicinchoninic acid (BCA) reagent (Promega, Madison, WI, USA). An iST Sample Preparation Kit (PreOmics, Martinsried, Germany) was used for protein zymolysis according to the manufacturer’s protocol. The peptide mixture was re-dissolved in buffer A (buffer A: H_2_O, pH 10.0, adjusted with ammonium hydroxide) and fractionated at high pH using a nanoACQUITY UPLC system (Waters Corporation, Milford, MA, USA) connected to a reverse-phase column (OSfLC250, Shanghai Omicsolution Co., Ltd., Shanghai, China). The column flow rate was maintained at 2 µL/min and the temperature maintained at 30 °C. An EasyPept Frac NANO automatic fraction collection system (OSAP0003, Shanghai Omicsolution Co., Ltd.) was used to collect the X fraction, and each fraction was dried in a vacuum concentrator for DDA: nano-HPLC–MS/MS analysis. For the DDA: nano-HPLC–MS/MS analysis, an UltiMate 3000 (Thermo Fisher Scientific, Waltham, MA, USA) liquid chromatography system was connected to a timsTOF Pro2 (Bruker Daltonics, Billerica, MA, USA). The instrument was operated in DDA PASEF mode with 10 PASEF scans per topN acquisition cycle and accumulation and ramp times of 100 ms each. For DIA: nano-HPLC–MS/MS analysis, the equipment and experimental process were like those used for DDA: nano-HPLC–MS/MS analysis. DIA data were acquired in the diaPASEF mode. Raw DIA data were analyzed using Spectronaut 18 (Biognosys AG, Zurich, Switzerland) with default settings. SpectronAut determined the ideal extraction window dynamically, depending on iRT calibration and gradient stability. The cutoff levels of the *q* value for the precursors and proteins were 1%. The PCA was performed to determine the differences in DEPs between the GBL and GCP groups. DEPs were selected with *p* value < 0.05 and Fold Change ≥ 2 or Fold Change ≤ 0.5. KEGG and GO for DEPs were analyzed between the GBL and GCP groups.

### 4.7. Combined Transcriptome and Proteome Analysis

The R (version 3.5.1) was used to analyze the correlation of transcriptomes and proteomes between the GCP and GBL groups. Maps with nine quadrants were created to illustrate alterations in gene and protein expression in the transcriptomic and proteomic data, with the maps showing the quantification and enrichment of genes or proteins in each region. Significant GO term or KEGG pathway was identified using the comparative analysis in transcriptome and proteome. The common DEGs were also analyzed in the common pathway to identify important genes and pathways between transcriptome and proteome.

### 4.8. Histological Analysis

Histological sections of the gills were prepared as Pan [38] reported. Gills in the GCP and GBL groups were fixed in 4% paraformaldehyde, then dehydrated by series of degraded alcohol and embedded in paraffin wax. Embedded gills were cut into sections (5 μm thick) using a microtome (Leica Microsystems, Ankara, Turkey). Cut sections were immersed in sequence in Environmentally Friendly Dewaxing Transparent Liquid I (Servicebio, Wuhan, China) for 20 min, Environmentally Friendly Dewaxing Transparent Liquid II (Servicebio, Wuhan, China) for 20 min, anhydrous ethanol I for 5 min, anhydrous ethanol II for 5 min, 75% ethyl alcohol for 5 min, and finally rinsed with tap water. The sections were then placed in a hematoxylin solution for 3–5 min and then rinsed with tap water. We then placed the sections, in sequence, in 85% ethanol for 5 min, 95% ethanol for 5 min, and eosin dye for 5 min. The sections were then placed in absolute ethanol I for 5 min, absolute ethanol II for 5 min, absolute ethanol III for 5 min, xylene I for 5 min, and xylene II for 5 min. All sections were mounted on glass slides with neutral gum. The stained sections were examined by light microscope (BX51).

## 5. Conclusions

In the present study, gill transcriptome, proteome, and histology were used to analyze the variations in copper-stressed *E. sinensis*. The combined analysis of transcriptomes and proteomes demonstrated that three DEGs overlapped in two omics. The comparative omics analysis indicated that seven GO terms were significantly enriched. Further analysis detected two common KEGG pathways. The histological analyses showed that gills in the GCP group had collapsed lamellae with the enlarged marginal vessel and shortened inter-lamellar space because of the disruption of the pillar cell and cuticle. This study demonstrated three omics results and clarified mechanisms of copper toxicity in Chinese mitten crab.

## Figures and Tables

**Figure 1 ijms-26-04711-f001:**
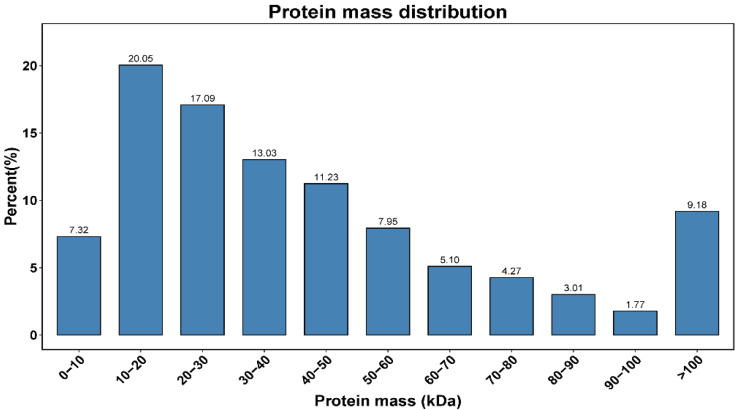
Protein mass distribution. The relative molecular masses of the identified protein were mostly distributed in the range of 10–100 kDa, with about 9.18% of relative molecular mass greater than 100 kDa.

**Figure 2 ijms-26-04711-f002:**
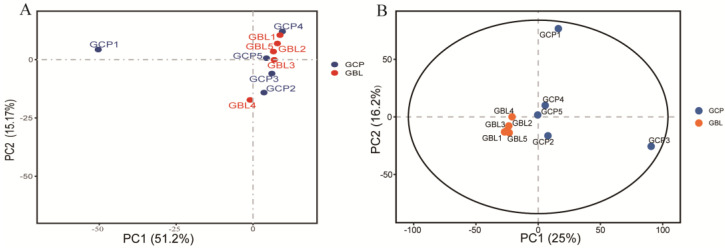
Principle component analysis of transcriptome (**A**) and proteome (**B**) abundance in the copper-stressed (GCP) and blank (GBL) groups. The orange plot indicates that the Chinese mitten crab was obtained from the GBL group; the blue plot indicates that the Chinese mitten crab was obtained from the GCP group.

**Figure 3 ijms-26-04711-f003:**
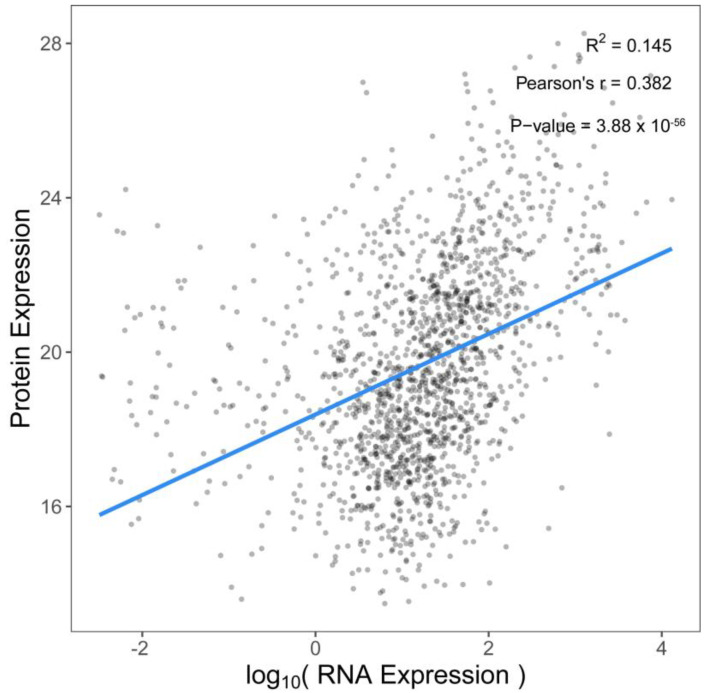
Correlation analysis between the transcriptome and proteome. A total of 1582 genes correlated with proteins. The Pearson correlation (blue line) between DEPs and DEGs was not high (R = 0.382).

**Figure 4 ijms-26-04711-f004:**
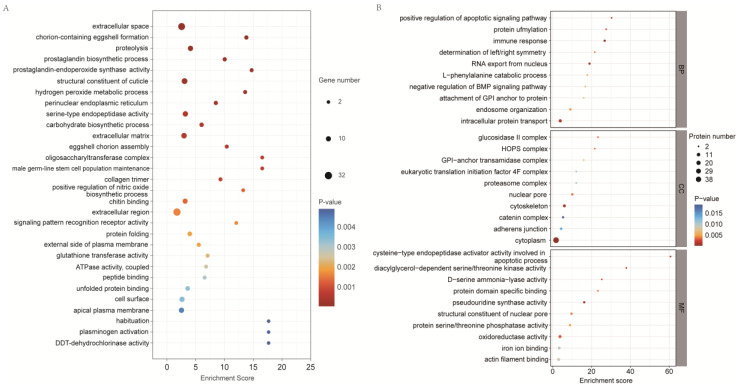
GO enrichment analysis of DEGs (**A**) and DEPs (**B**) between the copper-stressed (GCP) and blank (GBL) groups. Top 30 significantly enriched GO terms. (**A**) is the result of the GO enrichment annotation of DEGs, and (**B**) is result of the GO enrichment of DEPs.

**Figure 5 ijms-26-04711-f005:**
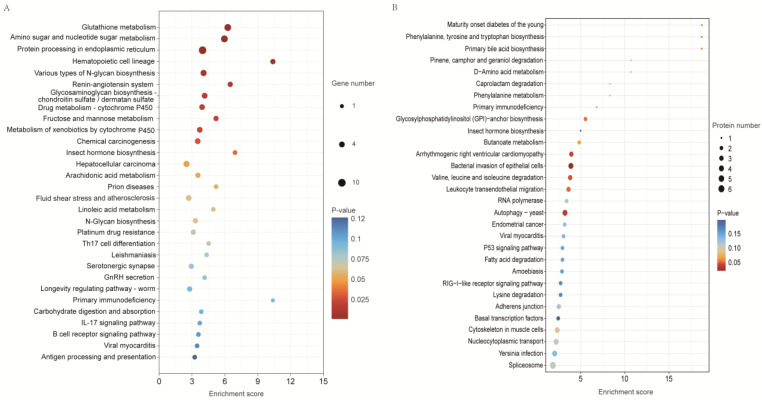
KEGG pathway analysis of DEGs (**A**) and DEPs (**B**) in the gills between the copper-stressed (GCP) and blank (GBL) groups. The x and y axes indicate the enrichment score and name of pathway, respectively.

**Figure 6 ijms-26-04711-f006:**
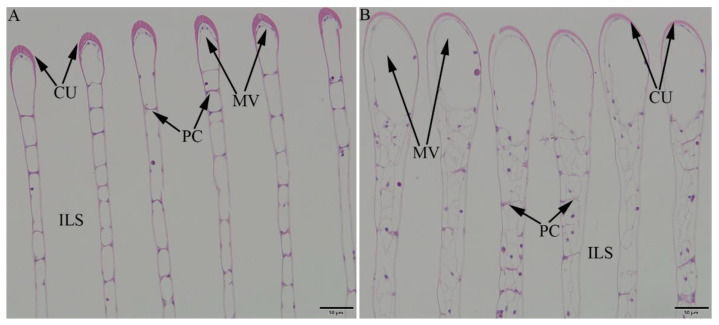
*Eriocheir sinensis* gill histology for the GBL (**A**) and GCP (**B**) groups. Cuticle (CU), pillar cell (PC), marginal vessel (MV), and inter-lamellar space (ILS). Bar markers = 50 μm in (**A**,**B**).

**Table 1 ijms-26-04711-t001:** *Eriocheir sinensis* transcriptome.

Reads Summary	GBL	GCP
Raw reads	241.35 M	235.66 M
Clean reads	236.92 M	230.56 M
Raw data	35.60 G	34.67 G
Clean data	34.95 G	33.91 G
Valid ratio (reads)/%	98.16	97.84
GC content %	46.98	46.53
Mapping reads ratio	66.75	72.29
Unique Mapped reads ratio	61.73	66.52

**Table 2 ijms-26-04711-t002:** Expression of the overlap DEGs in the transcriptome and proteome.

DGEs	Description	Transcriptome			Proteome		
		Log2FC	Regulation	*p* Value	Log2FC	Regulation	*p* Value
SACS	sacsin-like	−1.30	Down	0.030	5.89	Up	0.038
LOC126994049	minichromosome maintenance domain containing protein 2 like	3.47	Up	0.003	−11.79	Down	0.00001
stumps	DBB domain-containing protein stumps	1.70	Up	0.002	−6.96	Down	0.001

**Table 3 ijms-26-04711-t003:** GO terms significantly enriched by the overlapped DEGs in transcriptome and proteome.

GO ID	GO Term	GO Function	*p* Value	Gene ID
GO:0050727	regulation of inflammatory response	Biological process	0.003	stumps
GO:0014068	positive regulation of phosphatidylinositol 3-kinase signaling	Biological process	0.003	stumps
GO:0006270	DNA replication initiation	Biological process	0.004	LOC126994049
GO:0051897	positive regulation of protein kinase B signaling	Biological process	0.006	stumps
GO:0051321	meiotic cell cycle	Biological process	0.013	LOC126994049
GO:0006281	DNA repair	Biological process	0.034	LOC126994049
GO:0050750	low-density lipoprotein particle receptor binding	Molecular function	0.03	SACS

**Table 4 ijms-26-04711-t004:** Common KEGG pathways enriched by the overlapped DEGs.

Pathway	Pathway Name	*p* Value	Enrichment Score	Overlapped Genes
ko04662	B cell receptor signaling pathway	0.006	170.1	stumps
ko04151	PI3K-Akt signaling pathway	0.025	39.2	stumps

## Data Availability

All the data generated or used during the study appear in the submitted article.

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
