# Peer review of "Gill Transcriptome, Proteome, and Histology in Female *Eriocheir sinensis* Under Copper Stress"

_ijms, 2025, doi:10.3390/ijms26104711_

Round 1

Reviewer 1 Report

Comments and Suggestions for Authors

The paper (ijms-3607759) entitled "Gill transcriptome, proteome and histology in female Eriocheir sinensis under copper stress" investigated the transcriptomic profile, proteomic characterization, and morphological changes in the gill of Chinese mitten crab exposed to high level of copper. Results from this study could provide the basic data for further elucidating the toxic effects of copper on crab and its related molecular mechanisms.

Core content of this paper is valuable for monitoring the pollution situation of copper to protect of the resource of crustacean. However, the current paper was not well written, containing multiple grammatical and syntactic errors. The presentation of some sections in this manuscript, particularly discussion, are less clear and insufficient. Moreover, the results of RNA-seq and proteome in this study are based on bioinformatics analysis, lacking qRT-PCR/Western Blot verification.

Therefore, the authors should carefully re-check and modify the corresponding parts of this paper to improve its quality.

Major comments:

1.Regarding the "Abstract" part, some key information is missing, such as the initial size of Chinese mitten crab. Moreover, some descriptions in the current "Abstract" section had the awkward wording/ syntax and some ambiguity. For example, Line 8-10. It is suggested to reword these sentences for clarity.

2.Regarding the "1.Introduction" part, the original phrasing was inaccurate and contained multiple errors in syntax and format.

For example, the texts of Line 29, Line 30, Line 31, etc. had several language mistakes. Similarly, multiple long sentences contained grammatical and syntactical errors, which may reduce the readability of the manuscript or make readers confused about the meaning of the corresponding text. For example, Line 44-46, Line 51, Line 54-56, etc.

Some statements in the last paragraph of introduction contained the repeated descriptions. For example, the sentence of Line 69-70 partly repeats what was described in Line 66-67. It is suggested to compress or merge the corresponding texts.

Moreover, the relevant study regarding the negative influence of copper in aquatic animals is insufficient in the current introduction part. The corresponding information/background should be provided.

Thus, the authors should reword the relevant statement of "1.Introduction" for more clearly specifying background or significance in this study.

3.Multiple unnecessary and redundant descriptions are present in the original texts of "2.Results". For example, the sentences in Line 91-93, Line 141-143, etc, are speculative and discursive texts. It is suggested to move them to the discussion part.

Similarly, some sentences in Line 109, Line 173, etc, are the methodological descriptions. They could be removed directly or moved to the "4.Materials and Methods" part.

Some statements essentially repeat the same information on results, such as Line 165-166, etc.

Moreover, too many grammatical and syntax errors are present in "2.Results".

Thus, the main contents and the presentation of results in this section should be reworded.

4.The current texts of "3.Discussion" part were poorly written and need extensive proof reading for grammar and content. Multiple statements were repetitive of the results section. For example, the texts in Line 194-197.

Some contents regarding the previous study were simply stated, not fully compared with your study or provided the analysis/inference. For example, Line 186-188.

Moreover, the discussion is a bit short and it lacks depth in interpreting your results and comparing/analyzing the mechanism of copper stress between your study and relevant research in aquatic animals, such as the dosage effect of copper stress.

Additionally, there were many statement issues and grammatical errors.

Thus, it is strongly recommended that the authors rewrite and reorganize the whole discussion section for better clarifying your findings in this study.

5.Regarding the section of "4.Materials and Methods", some methodological descriptions are unclear.

During the acclimating period, what feed were used? Commercial diet or other feed? What about the proximate composition of this diet?

Only female crabs were used in this experiment? Why did not choose male ones?

Also, there are some mistakes in syntax and grammar.

Thus, the authors should add more relevant information. Please revise "4.Materials and Methods" section accordingly.

6.The current reference list is very chaotic, with some format errors, including old cited literatures, missing/wrong information on DOI, volume number and pagenumber, etc.

For example, in Reference 2, 14, 15, etc, the information on page number should be provided. Similarly, there was no volume number in Reference 2 and Reference 13.

The same journal is presented with different names in Reference 1 and Reference 33. "Aquatic Toxicology" or "AQUAT TOXICOL", which one is correct? In Reference 2, the information on journal name is incomplete. The names of co-authors in Reference 2 are also incomplete.

Moreover, the number of cited literatures published in 2020-2025 is 12, less than 30% of total references (total number of references: 39). Please make sure about 50% of the references are within 5 years (2020-2025).

Thus, authors should re-check the references format according to the related instructions and modify accordingly.

Minor comments:

1.The figure legend is missing or incomplete in several images. No information on the scale bar is provided in the legend of Figure 5.

Moreover, the clarity of some pictures in the article is not enough. For example, Figure 4 and Figure 5. The resolution of these pictures should be improved.

Other errors (highlighted in yellow) were marked in the PDF file.

So, this manuscript will be reconsidered after major revision.

Comments on the Quality of English Language

Multiple parts of this manuscript are poorly written, containing several awkward syntactic structure/phrasing and multiple grammatical issues.

It is strongly recommended that the text should be proofread by a professional or native speaker.

Author Response

Comments 1: Regarding the "Abstract" part, some key information is missing, such as the initial size of Chinese mitten crab. Moreover, some descriptions in the current "Abstract" section had the awkward wording/ syntax and some ambiguity. For example, Line 8-10. It is suggested to reword these sentences for clarity.

Response 1: The initial size of Chinese mitten crab has been added in the abstract.

Comments 2: Regarding the "1.Introduction" part, the original phrasing was inaccurate and contained multiple errors in syntax and format.

For example, the texts of Line 29, Line 30, Line 31, etc. had several language mistakes. Similarly, multiple long sentences contained grammatical and syntactical errors, which may reduce the readability of the manuscript or make readers confused about the meaning of the corresponding text. For example, Line 44-46, Line 51, Line 54-56, etc.

Response 2:  Line 29, Line 30, Line 31, Line 44-46, Line 51, and Line 54-56 have been corrected in the manuscript.

Comments 3: Some statements in the last paragraph of introduction contained the repeated descriptions. For example, the sentence of Line 69-70 partly repeats what was described in Line 66-67. It is suggested to compress or merge the corresponding texts.

Response 3: The repeated descriptions of histology has been merged in the text.

Comments 4: Moreover, the relevant study regarding the negative influence of copper in aquatic animals is insufficient in the current introduction part. The corresponding information/background should be provided.

Thus, the authors should reword the relevant statement of "1.Introduction" for more clearly specifying background or significance in this study.

Response 4: The relevant study regarding the negative influence of copper in aquatic animals has been added in the text.

Comments 5: Multiple unnecessary and redundant descriptions are present in the original texts of "2.Results". For example, the sentences in Line 91-93, Line 141-143, etc, are speculative and discursive texts. It is suggested to move them to the discussion part.

Response 5: Line 91-93 “Proteome PCA result (Figure 2B) indicated that samples between the GBL and GCP groups can be separated apparently.” has been moved to the discussion.

Line 141-143 “Which showed that these GO terms played important roles in E. sinensis once stimulated by copper.” has been moved to the discussion.

Comments 6: Similarly, some sentences in Line 109, Line 173, etc, are the methodological descriptions. They could be removed directly or moved to the "4.Materials and Methods" part.

Response 6: Line 109 “DEGs and DEPs with P < 0.05 were used for Gene Ontology (GO) analysis.” has been deleted.

Line 173 “Gills of the E. sinensis were collected at 96 h from GCP and GBL groups.” was also deleted.

Comments 7: Some statements essentially repeat the same information on results, such as Line 165-166, etc.

Response 7: Line 165-166 “A comparative analysis of transcriptomes and proteomes showed that 3 overlapped DEGs significantly regulated.” was deleted.

Comments 8: Moreover, too many grammatical and syntax errors are present in "2.Results".

Thus, the main contents and the presentation of results in this section should be reworded.

Response 8: Revisions have been made in the "2.Results".

Comments 9: The current texts of "3.Discussion" part were poorly written and need extensive proof reading for grammar and content. Multiple statements were repetitive of the results section. For example, the texts in Line 194-197.

Response 9: Line 194-197 “Expression of mRNA and protein in copper group and blank group was analyzed by transcriptome and proteome, respectively. Combined omics analysis showed that some genes and proteins had different expression levels between the GCP group and GBL group.” was deleted.

Comments 10: Some contents regarding the previous study were simply stated, not fully compared with your study or provided the analysis/inference. For example, Line 186-188.

Response 10: Line 186-188 “In order to investigate the role of Cu2+ on lipid metabolism, E. sinensis exposed to Cu2+ solution (50 μg/L) for four days, and gene of sterol regulatory element binding protein was knocked down by RNA interference” revised as “When E. sinensis exposed to Cu2+ solution (50 μg/L) for four days, expression of genes associated to lipid synthesis (ACC, FAS, SCD) is considerably reduced after a decline in SREBP expression. There are five genes down-regulated and two up-regulated in the lipid metabolism pathway in our study, indicating that lipid metabolism was inhibited under copper stress.”

Comments 11: Moreover, the discussion is a bit short and it lacks depth in interpreting your results and comparing/analyzing the mechanism of copper stress between your study and relevant research in aquatic animals, such as the dosage effect of copper stress.

Response 11: the dosage effect of copper stress has been added in the first paragraph of the discussion.

Comments 12: Additionally, there were many statement issues and grammatical errors.

Thus, it is strongly recommended that the authors rewrite and reorganize the whole discussion section for better clarifying your findings in this study.

Response 12: The discussion has been revised in the text.

Comments 13: 5.Regarding the section of "4.Materials and Methods", some methodological descriptions are unclear.

Response 13: The section of "4.Materials and Methods" has been revised in the text.

Comments 14: During the acclimating period, what feed were used? Commercial diet or other feed? What about the proximate composition of this diet?

Response 14: During the acclimating period, E. sinensis were fed with sinking pellets (Tongwei Feed Co., Ltd., Hefei, China) twice daily. The formula feed comprised 40% protein, 12.0% crude fiber, 4.0% lipids, 16.0% ash, 0.6% total phosphorus, and 2.2% lysine.

Comments 15: Only female crabs were used in this experiment? Why did not choose male ones?

Response 15: Female crabs are more valuable than male, so we did this experiment by female first. The toxicity of copper on male crab will be studied later.

Comments 16: Also, there are some mistakes in syntax and grammar.

Thus, the authors should add more relevant information. Please revise "4.Materials and Methods" section accordingly.

Response 16: "4.Materials and Methods" section has been revised as pointed out in the PDF version.

Comments 17: The current reference list is very chaotic, with some format errors, including old cited literatures, missing/wrong information on DOI, volume number and pagenumber, etc.

Response 17: All the references have been checked carefully.

Comments 18: For example, in Reference 2, 14, 15, etc, the information on page number should be provided. Similarly, there was no volume number in Reference 2 and Reference 13.

Response 18: All the references have been revised.

Comments 19: The same journal is presented with different names in Reference 1 and Reference 33. "Aquatic Toxicology" or "AQUAT TOXICOL", which one is correct? In Reference 2, the information on journal name is incomplete. The names of co-authors in Reference 2 are also incomplete.

Response 19: "Aquatic Toxicology" is correct. All the references has been checked carefully.

Comments 20: Moreover, the number of cited literatures published in 2020-2025 is 12, less than 30% of total references (total number of references: 39). Please make sure about 50% of the references are within 5 years (2020-2025).

Response 20: About 50% of the references are within 5 years (2020-2025).

Comments 21: Thus, authors should re-check the references format according to the related instructions and modify accordingly.

Response 21: All the references have checked and modified.

Comments 22: 1.The figure legend is missing or incomplete in several images. No information on the scale bar is provided in the legend of Figure 6.

Response 22: Figure legend have been added in Figure1, Figure1, Figure2, Figure3, Figure4, and Figure5. Scale bar has been added to the Figure 6.

Comments 23: Moreover, the clarity of some pictures in the article is not enough. For example, Figure 4 and Figure 5. The resolution of these pictures should be improved.

Response 23: The resolution of Figure 4 and Figure 5 have been improved from 300ppi to 400ppi.

Comments 24: Comments on the Quality of English Language

Multiple parts of this manuscript are poorly written, containing several awkward syntactic structure/phrasing and multiple grammatical issues.

It is strongly recommended that the text should be proofread by a professional or native speaker.

Response 24: The manuscript has been revised by an experienced English-speaking colleague.

Reviewer 2 Report

Comments and Suggestions for Authors

Brief summary: Transcriptome, proteome and histology were used on gill tissue for copper stewed Eriocheir sinensis. DEG and DEPs of gill -omics were analysed. combined analysis of these to omits revealed the common DEGs (differentially expressed genes. and DEPs (differentially expressed proteins) GO terms and KEGG pathways. Histology variations were also analysed in the gills. Results can clarify mechanisms of copper stress and stress in general - as three DEGs were overlapped in the two comics.

This is an interesting report, and I have few suggestions to improve it.

There is some potential to shorten the introduction. It is for example pretty obvious that gills are an essential organ in crustaceans (and any other higher aquatic animal)

The manuscript would benefit from a clearer distinction between There are some references to Methods in the Results section, ("sequenced by the Illumina sequence platform") this should be strictly placed in the Materials and Methods, which should be somewhat expanded.

The Discussion is short and adequate.

Author Response

Comments 1: There is some potential to shorten the introduction. It is for example pretty obvious that gills are an essential organ in crustaceans (and any other higher aquatic animal)

Response 1: “Gill is an essential organ in aquatic animals, and plays an important role in respiration, osmoreulation, excretion and ion transpor” has been revised as “Gill plays an important role in respiration, osmoreulation, excretion and ion transpor in aquatic animals”

Comments 2: The manuscript would benefit from a clearer distinction between There are some references to Methods in the Results section, ("sequenced by the Illumina sequence platform") this should be strictly placed in the Materials and Methods, which should be somewhat expanded.

Response 2: “Transcriptome was sequenced by Illumina sequence platform” were transfer from Results section to Methods section

Round 2

Reviewer 1 Report

Comments and Suggestions for Authors

The manuscript (ijms-3607759) entitled "Gill transcriptome, proteome, and histology in female Eriocheir sinensis under copper stress" has been revised as suggested. The authors reply to the reviewer’s comments one by one (online system).

The current revision is suitable for publication in your journal, though it still contains some minor errors.

For example, in Line 111-112, please indent the paragraphs for clarity.

The scientific name of species in the text of Line 205-Line 212 was non-italic. In Line 208 and Line 210, use "μg/L", not "ug/L".

In Reference 18, the information on page number is missing. Page number of Reference 16 should be presented as "118989", not "118989-".

Thus, it is recommended to accept this paper although there are some minor mistakes.